# The Importance of the Nephrologist in the Treatment of the Diuretic-Resistant Heart Failure

**DOI:** 10.3390/life13061328

**Published:** 2023-06-06

**Authors:** Ákos Géza Pethő, Mihály Tapolyai, Maria Browne, Tibor Fülöp, Petronella Orosz, Réka P. Szabó

**Affiliations:** 1Department of Internal Medicine and Oncology, Faculty of Medicine, Semmelweis University, 1085 Budapest, Hungary; 2Department of Nephrology, Szent Margit Kórhaz, 1032 Budapest, Hungary; mtapolyai@aol.com; 3Medicine Service, Ralph H. Jonson VA Medical Center, Charleston, SC 29401, USA; tiborfulop.nephro@gmail.com; 4Department of Medicine, Division of Nephrology, University of Maryland Medical Center, Baltimore, MD 21201, USA; mcbrowne.md@gmail.com; 5Medicine Service, Baltimore VA Medical Center, Baltimore, MD 21201, USA; 6Department of Medicine, Division of Nephrology, Medical University of South Carolina, Charleston, SC 29425, USA; 7Bethesda Children’s Hospital, 1146 Budapest, Hungary; orosz.petronella@bethesda.hu; 8Department of Pediatrics, Faculty of Medicine, University of Debrecen, 4032 Debrecen, Hungary; 9Department of Nephrology, Faculty of Medicine, University of Debrecen, 4032 Debrecen, Hungary; rpszabo@belklinika.com

**Keywords:** ascites, cardiorenal syndrome, diuretic resistance, mortality, nephrologist, heart failure, renin–angiotensin–aldosterone system, sodium–glucose cotransporter 2 inhibitor, quality of life

## Abstract

Heart failure is not only a global problem but also significantly limits the life prospects of these patients. The epidemiology and presentation of heart failure are intensively researched topics in cardiology. The risk factors leading to heart failure are well known; however, the real challenge is to provide effective treatments. A vicious cycle develops in heart failure of all etiologies, sooner or later compromising both cardiac and kidney functions simultaneously. This can explain the repeated hospital admissions due to decompensation and the significantly reduced quality of life. Moreover, diuretic-refractory heart failure represents a distinct challenge due to repeated hospital admissions and increased mortality. In our narrative review, we wanted to draw attention to nephrology treatment options for severe diuretic-resistant heart failure. The incremental value of peritoneal dialysis in severe heart failure and the feasibility of percutaneous peritoneal dialysis catheter insertion have been well known for many years. In contrast, the science and narrative of acute peritoneal dialysis in diuretic-resistant heart failure remains underrepresented. We believe that nephrologists are uniquely positioned to help these patients by providing acute peritoneal dialysis to reduce hospitalization dependency and increase their quality of life.

## 1. Introduction

Heart failure and cardiovascular diseases (CVDs) are the leading cause of death globally. The World Health Organization’s (WHO’s) declared goal is to reduce the number of premature deaths by 25% by 2025 through nine voluntary global targets [1]. The epidemiology and clinical manifestations of heart failure are intensively researched topics in cardiology. The risk factors leading to heart failure are well known; however, the current key challenge is the offer of effective treatments with sustained efficacy for these patients. A vicious cycle often develops in heart failure of different etiologies leading to progressive fluid accumulation and tissue hypoxia, ultimately being responsible for the disease’s poor prognosis through a simultaneous compromise of both kidney and heart functions. Repeated hospital admissions due to decompensation are the rule with a significantly reduced quality of life. Multidisciplinary treatment of patients with severe heart failure aims to stop or interrupt this downward spiral (Figure 1).

In treating heart failure, an effective fluid management program is crucial to reduce hospital readmissions and overall mortality. This is based on optimal diuretic use and other procedures. Therefore, cooperation with other professions, e.g., nephrologists, is often necessary to maintain euvolemia, but patient education is crucial and contributes to treatment success [2]. According to the latest guidelines of the European Society of Cardiology (ESC), heart failure can be classified into three groups on the basis of the heart’s left-ventricular ejection fraction: HFrEF (“heart failure with reduced ejection fraction”; ≤40%), HFmrEF (“heart failure with mildly reduced ejection fraction”; 41–49%), and HFpEF (“heart failure with preserved ejection fraction”; ≥50%). This staging better defines therapeutic options and guidelines [3]. The current therapeutic approaches include basic drug regimens: angiotensin-converting enzyme inhibitors or angiotensin receptor–neprilysin inhibitors, mineralocorticoid receptor antagonists, beta-blockers, and the newest sodium–glucose cotransporter 2 inhibitor (SGLT2i) drug family, which also proves promising in the treatment of heart failure. Presently, two SGLT2 inhibitors, dapagliflozin and empagliflozin, have been demonstrated to exert beneficial effects on morbidity and mortality in HFrEF and HFpEF [4,5]. In addition to drug treatment options, many device therapy options can also help in the treatment of patients [6,7]. In moderate heart failure (HFmrFR and HFpEF), conservative treatments are the key therapeutic options. Diuretic resistance is a real challenge in decompensated heart failure. Diuretic resistance occurs when the therapeutically desired relief of congestion fails despite using an adequate diuretic dose. Such phenomena typically co-occur with acute or chronic kidney functional impairment.

Considering the above, we aimed to explore possible nephrological treatments for severe diuretic-resistant heart failure in our narrative review. Diuretic resistance and cardiorenal syndrome are conditions where special nephrological therapies may be needed to influence the disease’s outcome or improve the patient’s quality of life.

## 2. Search Methods

We searched the PubMed database and Google Scholar using the keywords “heart failure”, “diuretic resistance”, “diuretic-resistant heart failure”, “diuretic-resistant heart failure treatment”, “cardiorenal syndrome”, and “nephrologist” and their combinations. We followed the guidelines for literature search and manuscript preparation [8,9]. The inclusion criteria comprised all original articles (human studies only) and systematic reviews published until 1 April 2023. We did not specify the start time in the search; we only used the first publication available for the given keyword as a basis. We excluded repeated publications. Pertinent publications were located by looking through the references of retrieved articles (backward search) and more recent articles that cited the recovered paper (ahead search).

Studies addressing the objectives of the review and our hypotheses were extracted. Data collection for each study focused primarily on possible connections to our theory. Three authors independently screened each retrieved article for eligibility and extracted study data. Any discrepancies between the first and senior authors (Á.G.P., M.T., and T.F.) were resolved by discussion and agreement.

## 3. Results

On the basis of the above, in our narrative review statement, we wanted to draw attention to the possibility of nephrological treatments for severe diuretic-resistant heart failure. The most studied connection was “heart failure” and “nephrologist”. The announcements corresponding to the search terms are listed in Table 1. It can be seen from the listed publications that the association between “heart failure” and other diseases has been a concern of researchers since 1879.

Not surprisingly, the epidemiology, molecular biology, symptomatology, and cardiology of heart failure have been intensively researched and published. In the same way, a very significant number of publications have been published on the topic of diuretic resistance. Unfortunately, however, only 15 publications have been published on intertwining diuretic resistance and nephrology. Furthermore, researchers have dealt with cardiorenal syndrome since 1946, but reports on its connection with nephrology have only been published since 2002. All these results draw attention to the fact that nephrological therapeutic options are not considered in treating certain cardiology diseases.

## 4. Diuretic Resistance to Heart Failure

Heart failure, including diuretic-resistant heart failure, can be responsible for the repeated hospitalization of such patients. Unfortunately, epidemiological data show that this number is regrettably increasing [10]. In treating fluid overload in severe heart failure, loop diuretics are the mainstay of diuretic treatment. Loop diuretics reversibly inhibit sodium reabsorption in the thick ascending limb of the loop of Henle. This is why loop diuretics are primarily used, since most sodium reabsorption occurs in the ascending limb, enabling a more significant diuresis.

Other diuretics, such as thiazides, act on the distal ascending limb, while potassium-sparing diuretics work on the collecting ducts. Sodium reabsorption is lower in those sections of the nephron [11]. The family of loop diuretics, such as the most used furosemide, torasemide, and bumetamide, theoretically inhibits the same symporter in the apical membrane of the macula densa cells. Reduced sodium reabsorption further inhibits tubuloglomerular feedback and, thus, stimulates renin secretion [12]. On the other hand, the renin–angiotensin–aldosterone system (RAAS) increased by loop diuretics increases the plasma renin level, which also causes an increase in angiotensin II.

Nevertheless, this increase in RAAS activity can further inhibit tubuloglomerular feedback, thus reducing the glomerular filtration rate (GFR), i.e., a narrowing of kidney function [12]. RAAS activated by loop diuretics increases systemic blood pressure, as well as sodium and water reabsorption. In addition, loop diuretics also increase the level of vasodilator prostaglandins, as a result of which an increase in pressure also occurs in the proximal tubule [13]. Due to the countervailing effects of these biochemical processes, high doses of intravenous (IV) loop diuretics can either decrease or increase arterial blood pressure. Furthermore, blood flow to the kidneys decreases due to centralizing circulation [14], and decreasing kidney function itself further reduces the effective fluid removal with diuretics.

Another reason for diuretic resistance, in addition to the activation of the RAAS, is that structural changes in the nephrons also occur, also known as the “braking phenomenon”. Chronic use of loop diuretics results in hypertrophy and hyperplasia of the cells of the distal tubules. Hypertrophy of distal tubular cells reduces the response to diuretics due to compensatory increased sodium reabsorption [15].

SGLT2 inhibitors are a promising new drug family in treating heart failure. The beneficial effects of SGLT2 inhibitors on heart failure were also confirmed by four large randomized controlled trials (DAPA-HF, SOLOIST-WHF EMPEROR-Reduced, and EMPEROR-Preserved), in which a total of 15,684 patients participated. Studies have shown that using SGLT2i reduces the number of cardiovascular deaths and the hospitalization rate of patients with heart failure. The beneficial effect of SGLT2 inhibitors on heart failure was independent of left-ventricular ejection fraction (LVEF) status [16]. In heart failure patients with mildly reduced or preserved ejection fraction, dapagliflozin reduces the risk of the primary composite outcome (DELIVER-HF) [17]. The EMPULSE study showed that the initiation of empagliflozin as part of usual care in patients who are hospitalized for acute heart failure will result in a clinically meaningful benefit in 90 days without safety concerns [18]. SGLT2 inhibitors are mainly used in type 2 diabetes to control plasma glucose. However, inhibition of the sodium–glucose cotransporter (SGLT) 2 in the proximal tubule of the kidney has many additional effects on kidney function, plasma volume homeostasis, and energy metabolism throughout the body. By inhibiting SGLT-2-dependent glucose and sodium reabsorption, in addition to glucosuria, natriuresis also increases, as a result of which the distal tubular sodium load increases. Therefore, in addition to glucosuria, it creates net effective natriuresis [19,20,21,22]. The commonly used diuretics are listed in Table 2.

Another interesting therapeutic option is that hypertonic solutions could help to break through the diuretic resistance in decompensated heart failure [23]. The explanation for this simple therapeutic intervention is that long-term use of furosemide leads to hyponatremia and hypochloremia. This directly leads to diuretic resistance. Therefore, intermittent and intravenous administration of furosemide may be recommended in acute decompensation [24].

## 5. Cardiorenal Syndrome

As discussed above, elevated serum renin levels raise intraglomerular pressure due to efferent arteriolar constriction. In severe, decompensated heart failure, due to significantly increased renal venous pressure and reduced renal blood flow, the compensation that preserves GFR fails, thus resulting in a further decrease in GFR [25]. The progression of cardiorenal syndrome into more and more severe heart failure is, thus, the result of not only sodium accumulation but perhaps also an increase in the intra-abdominal compartment [26]. The fact that hypertonic saline infusion with furosemide may be more effective than diuretics only [27] and the phenomenon that a normal-sodium diet compared to a low-sodium diet improves outcome [28] all point our attention to reconsider sodium excess as the only instigator of cardiac decompensation. As a result of RAAS and neurohumoral activation, preglomerular vasoconstriction results in decreased intraglomerular pressure and reduced GFR. To maintain an adequate plasma volume, increased activation of the neurohumoral axis results in increased proximal tubular sodium and water reabsorption, ultimately resulting in oliguria and worsening congestion [29].

However, hemodynamic changes can lead to cardiorenal syndrome (CRS) and the activation of the sympathetic nervous system, through persistent RAAS activation. This also includes chronic inflammation and an imbalance in the ratio of reactive oxygen species/nitrogen monoxide production [30,31]. Specific circulating cytokines can be responsible for adverse cardiac side-effects in both acute and chronic kidney damage. In acute kidney injury, IL-6 (interleukin-6), TNF-α (tumor necrosis factor-α), and IL-1 (interleukin-1) have a cardio-depressant direct impact. The role of FGF-23 (fibroblast growth factor-23) in uremic cardiomyopathy (type 4 CRS) in chronic kidney damage is also known [32]. Fluid overload and consequent peripheral venous congestion in severe heart failure result in endothelial dysfunction and additional proinflammatory cytokines released by endothelial cells. This increased proinflammatory cytokine production also plays a role in developing CRS and changes in hemodynamic parameters [33]. The presence of abdominal ascites is known to pose an additional burden on net renal perfusion [34,35]. This also shows how complex the pathophysiology of CRS is and how closely related the heart and kidney functions are (Figure 2).

Recognizing a broader clinical spectrum that may represent cardiorenal dysregulation, the Acute Dialysis Quality Initiative outlined a consensus approach in 2008 that phenotypes CRS into two main groups. Isolation was proposed to be defined on the basis of the primary disease process in the development of CRS; thus, we can differentiate between cardiorenal and renocardiac syndromes [36]. The purpose of this consensus definition of CRS is to facilitate reliable characterization of the clinical manifestations of cardiorenal disorders for diagnostic and therapeutic purposes, simplify inclusion criteria in epidemiological studies, identify target populations for treatment, and develop new diagnostic tools for the diagnosis of CRS and management (Table 3).

## 6. Peritoneal Dialysis in Diuretic-Resistant Heart Failure

Peritoneal dialysis (PD) treatment in severe diuretic-resistant heart failure can improve a patient’s quality of life and functional class stage of heart failure; however, cardiovascular mortality remains high [37]. The first published report of PD treatment offered for congestive heart failure is dated back to 1996 [38]. PD treatment is the gentlest procedure in such patients to reduce or eliminate fluid overload. These patients are most often readmitted to the hospital due to fluid overload, and PD treatment can effectively remove excess fluid and salt according to the details above [39]. Furthermore, it also prevents progressive ascites accumulation, known to impair net renal perfusion pressure. In the case of a cardiorenal syndrome associated with severe heart failure (CRS I-II), the degree of kidney damage does not affect survival [40]. However, improving left-ventricular systolic function in patients with PD treatment may result in a better quality of life [41,42]. It is also known that PD treatment can only effectively improve the quality of life of patients with severe LVEF who have markedly reduced left-ventricular systolic function (EF < 15%) [41]. The improvement in quality of life can be dramatic. PD therapy can improve the functional status of patients with severe diuretic-resistant heart failure whose vital functions can only be maintained by vasopressors [43]. Several reports have reported decreased hospitalization with PD treatment in such patients [44,45]. In addition to the significant reduction in the need for hospitalization in the elderly patient group, the indisputable advantage of PD treatment over alternative extracorporeal treatment is that it is a more cost-effective procedure [46,47]. In such a group of patients, PD treatment is preferable to hemodialysis (HD) treatment, among other reasons, because, in global heart failure caused by right-heart failure, ascites due to cardiorenal syndrome also occur due to stagnation of the portal circulation. As the peritoneal membrane does not generate an osmotic or sodium gradient, net ascites removal always implies isotonic salt–water extractions, whereas net ultrafiltration on PD may be slightly hypotonic [48,49].

In a retrospective study of nearly 11,000 patients, repeated hospitalizations of patients treated with hemodialysis (HD) and PD who required renal replacement therapy due to severe heart failure were compared. The study showed that the incidence of PD peritonitis and the resulting hospitalization rate were significantly higher in patients treated with PD. However, PD treatment still seemed more beneficial [50]. Despite a higher incidence of rehospitalization due to PD peritonitis, PD treatment is a more favorable renal replacement modality for severe diuretic-resistant heart failure than HD treatment. In a retrospective study of nearly 10,000 patients admitted with manifest heart failure, the group treated with HD was readmitted to the hospital significantly more times than the group treated with PD [45].

PD treatment in a group of patients suffering from severe diuretic-resistant heart failure has been known for several decades. However, it has not entered the everyday public consciousness, and no professional consensus has yet been formed. It has been shown to effectively reduce the need for hospital admissions and improve left-ventricular systolic function in several studies. In the case of associated kidney damage, the results were not favorable; in the CARESS HF study (Effectiveness of Ultrafiltration in Treating People with Acute Decompensated Heart Failure and Cardiorenal Syndrome), kidney function did not improve significantly, although, from the point of view of the outcome, the aim was not necessarily to improve kidney function [51]. The latest recommendation of the AHA (American Heart Association) and the ESC (European Society of Cardiology) based on the results of the CARESS study recommends ultrafiltration with IIb evidence only in the group of patients with severe diuretic-resistant heart failure [52].

The mortality of patients with diuretic-resistant heart failure is very high; thus, extracorporeal treatment (HD) performed with a central venous catheter (CVK) represents an additional increased risk due to bloodstream infection and death [53]. In the presence of other indwelling intravascular CV hardware, such as pacemakers or ventricular synchronizers, the risk of bacteremia is even higher with CVK [54,55]. If renal replacement therapy is justified, PD treatment can particularly benefit severe diuretic-resistant heart failure patients [56,57]. In addition, patients with diuretic-resistant heart failure are often severely malnourished; during PD treatment, the glucose absorbed from the solution can improve the nutritional status and reduce the severity of cardiac cachexia [58].

Atrial natriuretic peptide (ANP) and tumor necrosis factor-α (TNF-α) were removed during PD treatment in heart failure despite no improvement in renal function. In those patients with PD treatment, removed interleukin (IL)-1 and IL-6 also played a role in the functional phase and increased measurable ejection fraction [59]. The effectiveness of PD treatment depends not only on the transport of low-molecular-weight substances but also on uremic toxins secreted into the abdominal cavity. In this group of patients, the transperitoneal transport mechanism has additional potential benefits; by removing inflammatory and other cardiotoxic molecules, it contributes to an increase in myocardial contractility [60].

We believe that the cardiotoxic molecules removed during PD treatment also contribute to improving cardiac status due to salt and fluid removal (see cardiorenal syndrome briefly discussed above).

## 7. Urgent Start of the Peritoneal Dialysis in Diuretic Resistance Heart Failure

Minimally invasive percutaneous PD catheter insertion has been a procedure described for decades. Of course, there are cases where surgical PD catheter (PDC) insertion is indispensable, e.g., in overweight, obese patients or individuals who have undergone major abdominal surgery, in whom there is a possibility of intra-abdominal adhesions [61]. In the case of suspicion of abdominal adhesions, a laparoscopic intervention is recommended, during which any adhesions can be surgically dissolved, and the catheter can be inserted into the appropriate place; alternatively, if placement is not feasible, the procedure can be aborted. In the latest international recommendations (ISPD—International Society for Peritoneal Dialysis), the point of the passage of the percutaneous PD catheter through the abdominal wall is not strictly defined. When inserting a percutaneous PDC, the most important thing is the positioning of the catheter; therefore, the end of the catheter must be located in the pelvis, and the catheter exit site must be at least 2 cm in the abdominal wall from the outer Dacron ring [62]. The ISPD recommendation considers a waiting period of at least 2 weeks after the insertion of the PD catheter to be desirable. However, the recommendation also acknowledges that, in several studies, no significant difference was found with regard to PD peritonitis or modality survival in patient groups in which PD–fluid exchanges had to be started urgently after the intervention [63,64,65]. Only a few leakages were observed during PD treatment initiated with an emergency indication in clinical trials [66].

Recently, PD treatment with an emergency indication has been increasingly considered in treating severe septic conditions and acute kidney damage caused by the pandemic SARS-CoV-2 virus infection. Given the limited possibilities of extracorporeal treatment and those inflammatory markers—such as IL-6—being effectively removed with PD treatment, the cytokine storm caused by the COVID-19 infection can be mitigated [67,68]. It is also obvious that, with due care, even “bedside” percutaneous PD catheter insertion and emergency PD treatment starting within 24 h after the intervention does not pose an increased risk of possible technical complications. Of course, the ideal case is to wait for the catheter to settle correctly. Nevertheless, in the event of an emergency indication, the minimally invasive procedure also offers the possibility of symptomatic treatment of acute kidney damage [69,70].

With the increasing acceptance of minimally invasive procedures, percutaneous PDC can be considered in patients with severe, critical circulatory failure under regional anesthesia. Most severely critically ill patients may already be sedated unconscious due to acute critical illness [44]. Percutaneous PD implantation and, in the case of emergency, the initiation of PD treatment are, in principle, comparable to the insertion of a tunneled dialysis catheter (TDC), with subsequent imminent use after position confirmation. Various abdominal injuries and early infections are possible complications, which are much lower when using the percutaneous technique [71]. According to some studies, no significant difference was found between the two procedures in terms of catheter dysfunction or leakage of PD solution. However, the incidence of PD peritonitis was significantly lower in the percutaneous group [72]. In the outcome of PD catheter implantation success, there was no statistical difference in survival between surgical and percutaneous placement [73]. In our case series, with the follow-up of our patients, we demonstrated no PDC placement-related complication and robust modality retention [74]. Our group successfully utilized a Veress needle (spring-loaded needle to reduce internal injury) [75] to perform acute PD treatment. It is even more interesting that we performed acute PD treatment in two of our patients with severe heart failure awaiting heart transplants [76]. It should be pointed out that the presence of cardiac ascites makes PDC catheter insertion only easier, as the presence of fluid would make intrabdominal wire placement safer; furthermore, an ‘early-start’ PD would prevent recurring ascites accumulation and achieve vigorous salt–water removal at the same time. PD treatment started after percutaneous PD catheter insertion can be an effective alternative to HD treatment in patients with severe heart failure [77].

## 8. Other Specific Nephrology Treatments in Diuretic-Resistant Heart Failure

One major comorbidity to modulate risk is the presence of anemia, which emerges with both heart failure and kidney diseases. In heart failure, anemia also correlates with the functional stage of severe diuretic-resistant heart failure and the degree of clinical symptoms [78]. It follows that treating anemia of clear renal origin can also be a therapeutic goal in such patients. Furthermore, treating the associated anemia clearly improves the heart’s pump function [79].

The presence of iron deficiency had a negative prognostic impact only in HFrEF but not in HFpEF. Persistent iron deficiency was strongly associated with mortality after a 6 month follow-up [80]. Several intravenous (IV) iron products have been used in HF research; however, ESC guidelines currently recommend only iron carboxymaltose [81,82]. This recommendation may change following the recent publication of the IRONMAN study, which showed benefits in HFrEF patients using ferric derisomaltose [83]. A recent meta-analysis showed that intravenous iron carbohydrate therapy significantly reduced hospitalizations for worsening heart failure and first hospitalizations for worsening heart failure or death but did not significantly affect all-cause mortality. No significant differences in side-effects were observed between treatment groups [82]. A retrospective database analysis showed that intravenous iron appears to improve ejection fraction and cardiac functional status in outpatients with iron deficiency, HFpEF, and HfrEF [84].

## 9. Discussion

As discussed above, a close relationship exists between heart and kidney functions. In this narrative review, our goal was to highlight that, in diuretic-resistant heart failure, PDC placement by a nephrologist or intervention specialist would enable acute PD and efficacious immediate control of fluid overload. It is critical to understand that this procedure is not curative for heart failure but intended to improve the patient’s quality of life and is vital for successful rehabilitation. Indeed, in the opinion of these writers, what is not well understood is when the *ideal* time point of intervention would be to address cardiorenal failure with PD, rather than awaiting the point of absolute necessity. As we have shown, few publications specifically dealt with the correlations between diuretic-resistant heart failure and the possibilities of nephrological treatments. However, the number of experiences and reports obtained by combining these procedures is negligible; few researchers have dealt with this possibility so far.

We believe that, if conservative treatment options are exhausted in diuretic-resistant heart failure and extracorporeal ultrafiltration becomes necessary, in the absence of contraindications, PDC placement and acute PD treatment should be the preferential treatment to prevent repeated and long-term hospital admissions and to treat the development of fluid overload. The results of the publications published on this topic are encouraging, but prospective studies on many patients are necessary to verify our hypothesis. Furthermore, when planning prospective studies, we must also keep in mind the change in the quality of life since this represents a major outcome in this patient group.

## 10. Conclusions

From the presented reports, we can confirm that, in the treatment of severe diuretic-resistant heart failure, acute nephrological procedures are underrepresented as acute PD treatment. Especially in cases with ascites, when this becomes necessary due to hypervolemia, it is recommended to consider acute PD treatment. The strength of our hypothesis and the narrative reviews is that both PD treatment in heart failure and PDC placement have been intensively researched and published for decades, and numerous publications have reported on the effectiveness of the procedures. The limitation of our hypothesis is that there are only a few publications about these involving patients.

## Figures and Tables

**Figure 1 life-13-01328-f001:**
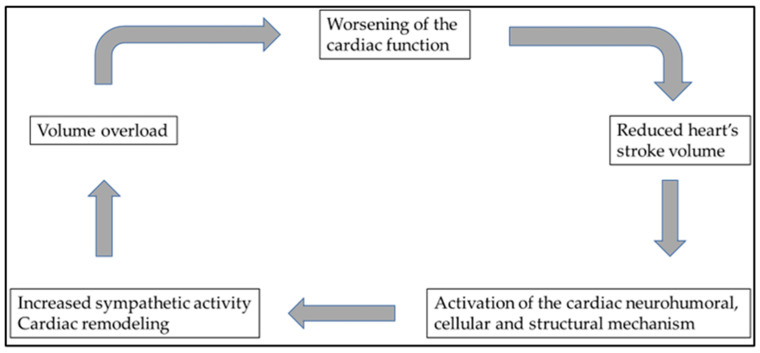
Multidisciplinary treatment of patients with severe heart failure aims to stop or interrupt this downward spiral. Unfortunately, heart failure of various etiologies eventually ends up in this vicious cycle, ultimately responsible for the disease’s poor prognosis.

**Figure 2 life-13-01328-f002:**
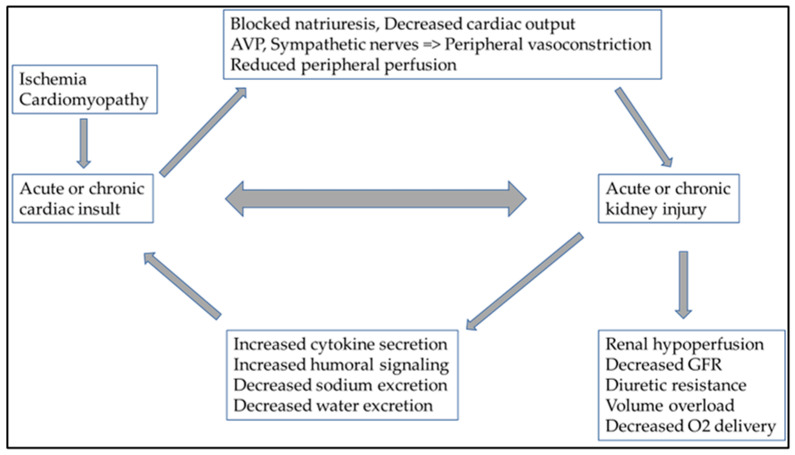
In cardiorenal syndrome, heart and kidney function are closely related, as are the pathophysiologies of several neurohumoral and inflammatory pathways. Ultimately, the deterioration of either the heart or kidney function causes consequential damage to the other organ. Abbreviations: AVP: arginine vasopressin; GFR: glomerular filtration ratio; O_2_: oxygen.

**Table 1 life-13-01328-t001:** The search results for the keywords “heart failure”, “diuretic resistance”, “diuretic-resistant heart failure”, “diuretic-resistant heart failure treatment”, ”cardiorenal syndrome”, and “nephrologist”.

Searching Keywords	Number of Publications until 1 April 2023	First Publication in This Field
“heart failure”	309,020	1879
“heart failure” and “nephrologist”	352	1983
“diuretic resistance”	7087	1952
“diuretic resistance” and “nephrologist”	22	1976
“diuretic-resistant heart failure”	1361	1952
“diuretic-resistant heart failure” and “nephrologist”	15	1996
“diuretic-resistant heart failure treatment”	2	1994
“diuretic-resistant heart failure treatment” and “nephrologist”	0	-
“cardiorenal syndrome”	2322	1946
“cardiorenal syndrome” and “nephrologist”	66	2002

**Table 2 life-13-01328-t002:** Possible used diuretics in severe heart failure.

Diuretics	Name	Effect on Nephron	Dose	Mechanism of Action	Overall Outcome
Loop diuretics	Furosemide, bumetanide, torsemide	Henle loop	40 mg furosemide is equivalent to 1 mg bumetanide and 20 mg torsemide	Na^+^–K^+^–2Cl^−^ symporter	Decreasing hypertonic renal medulla
Thiazides	Hydrochlorothiazide, chlortalidone	Distal convoluted tubule	1−2 × 50 mg/day	Na^+^–Cl^−^ transporter	Releasing NaCl and water
Aldosterone antagonist	Spironolactone	Distal tubule	100 mg/day	Na^+^/K^+^ ATPase	Increasing sodium excretion
Carboanhyd-rase inhibitor	Acetazolamide, methazolamide	Proximal convoluted tubules	Typically, 250 mg/day	Causing the accumulation of carbonic acid	Increasing urinary Na^+^ and bicarbonate (HCO_3_^−^)
SGLT2 inhibitors	Dapagliflozin, empagliflozin	Proximal tubules	Typically, 10 mg/day	Sodium–glucose transport proteins	Glucosuria and natriuresis

Abbreviations: ATPase: adenosine 5′-triphosphatase; HCO_3_^−^: hydrogen carbonate anion; mg: milligram; NaCl: sodium chloride; SGLT2: sodium–glucose cotransporter 2.

**Table 3 life-13-01328-t003:** Classification of cardiorenal syndrome based on the consensus conference of the acute dialysis quality initiative.

CRS Type	Nomenclature	Description	Clinical Examples
CRS 1	Acute CRS	HF resulting in AKI	ACS resulting in cardiogenic shock and AKI; AHF resulting in AKI
CRS 2	Chronic CRS	Chronic HF resulting in CKD	Chronic HF
CRS 3	Acute renocardiac syndrome	AKI resulting in AHF	HF in the setting of AKI from volume overload, inflammatory surge, and metabolic disturbances in uremia
CRS 4	Chronic renocardiac syndrome	CKD resulting in chronic HF	LVH and HF from CKD-associated cardiomyopathy
CRS 5	Secondary CRS	The systemic process resulting in HF and kidney failure	Sepsis, liver cirrhosis, and amyloidosis

Abbreviations: ACS: acute coronary syndrome; AHF: acute heart failure; AKI: acute kidney injury; CKD: chronic kidney disease; CRS: cardiorenal syndrome; HF: heart failure; LVH: left-ventricular hypertrophy.

## Data Availability

Not applicable.

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
