# Peer review of "The Importance of the Nephrologist in the Treatment of the Diuretic-Resistant Heart Failure"

_life, 2023, doi:10.3390/life13061328_

Round 1
Reviewer 1 Report
Pethő et al., wrote nice review on the role of nephrologist for the treatment of diuretic resistant heart failure. For this, nephron-physicians provided acute peritoneal dialysis to those patient, which may lower hospital burden and increase their quality of life.
These kinds of works are exceptional to appreciate the work of nephrologist.
Overall, this is very interesting work and I recommend this for publication.
Author Response
Dear Reviewer,
On behalf of my co-authors, thank you very much for your positive opinion.
Kind regards,
Ákos Pethő, MD,PhD
Reviewer 2 Report
The manuscript presented by A. G. Petho et al., illustrates the vital role played by acute peritoneal dialysis in helping patients with diuretic resistance heart failure. The following manuscript is well assembled review with a focus on the importance of nephrologists in improving the quality of life in these patients. However following points needs revision from the authors: -
1. Kindly rearrange the language in line 68 to give a clearer meaning.
2. Another innovative approach applied by nephrologists in diuretic resistance cardio-renal syndrome patients was the intravenous administration of loop diuretics in combination with other classes of diuretics. To better update the discussion, authors could refer to and cite “Diuretic Resistance in Cardio-Nephrology: Role of Pharmacokinetics, Hypochloremia, and Kidney Remodeling. Kidney Blood Press Res. 2019;44(5):915-927.https://doi.org/10.1159/000502648”
3. In table 2, the authors could mention and add 2 more columns saying, “mechanism of action” (for example, inhibiting sodium reabsorption) and “Overall outcome” (for example, increased RAAS mechanism, plasma renin levels, and angiotensin II) to mention the way these diuretics work and their ultimate effects.
4. Kindly mention that Cardiorenal syndrome is denoted by the acronym CRS.
5. Please go through the figures submitted in the manuscript to check whether they have equal proportions and dimensions for the arrows and boxes used.
6. Authors need to mention “HD” refers to “hemodialysis” when mentioning it for the first time in line 256.
The quality of English presented in the manuscript is understandable and reader-friendly. some minor changes are required
Author Response
Dear Reviewer,
On behalf of my co-authors, thank you very much for your positive opinion. We made the recommended improvement listed below.
- Kindly rearrange the language in line 68 to give a clearer meaning.
Answer: Thank you, I changed this sentence for a better meaning.
- Another innovative approach applied by nephrologists in diuretic resistance cardio-renal syndrome patients was the intravenous administration of loop diuretics in combination with other classes of diuretics. To better update the discussion, authors could refer to and cite “Diuretic Resistance in Cardio-Nephrology: Role of Pharmacokinetics, Hypochloremia, and Kidney Remodeling. Kidney Blood Press Res. 2019;44(5):915-927.https://doi.org/10.1159/000502648”
Answer: Thank you for the advised reference; I added this to the manuscript with a brief explanation.
- In table 2, the authors could mention and add 2 more columns saying, “mechanism of action” (for example, inhibiting sodium reabsorption) and “Overall outcome” (for example, increased RAAS mechanism, plasma renin levels, and angiotensin II) to mention the way these diuretics work and their ultimate effects.
Answer: Thank you for your advice. I have added the recommended columns to the table.
- Kindly mention that Cardiorenal syndrome is denoted by the acronym CRS.
Answer: Thank you, I explained the meaning of the abbreviation
- Please go through the figures submitted in the manuscript to check whether they have equal proportions and dimensions for the arrows and boxes used.
Answer: Thank you. I reviewed the figures and standardized the dimensions of the arrows.
- Authors need to mention “HD” refers to “hemodialysis” when mentioning it for the first time in line 256.
Answer: Thank you, I added the meaning of “HD”.
Kind regards,
Ákos Pethő, MD, PhD